META-RESEARCH

# How significant are the public dimensions of faculty work in review, promotion and tenure documents?

**Abstract** Much of the work done by faculty at both public and private universities has significant public dimensions: it is often paid for by public funds; it is often aimed at serving the public good; and it is often subject to public evaluation. To understand how the public dimensions of faculty work are valued, we analyzed review, promotion, and tenure documents from a representative sample of 129 universities in the US and Canada. Terms and concepts related to public and community are mentioned in a large portion of documents, but mostly in ways that relate to service, which is an undervalued aspect of academic careers. Moreover, the documents make significant mention of traditional research outputs and citation-based metrics: however, such outputs and metrics reward faculty work targeted to academics, and often disregard the public dimensions. Institutions that seek to embody their public mission could therefore work towards changing how faculty work is assessed and incentivized.
DOI: https://doi.org/10.7554/eLife.42254.001

**JUAN P ALPERIN\*, CAROL MUÑOZ NIEVES, LESLEY A SCHIMANSKI, GUSTAVO E FISCHMAN, MEREDITH T NILES AND ERIN C MCKIERNAN**

## Introduction

Review, promotion and tenure (RPT) processes are a cornerstone of academic life at higher education institutions in the United States and Canada. They can influence where faculty focus their attention, the activities they choose to pursue, and choices such as the direction of their research program and the venues where they publish their work, especially during the pre-tenure period (*Harley et al., 2010*). Unsurprisingly, RPT has been the subject of much scrutiny (for examples see *Gordon, 2008*; *Schimanski and Alperin, 2018*). While previous studies (*Gardner and Veliz, 2014*; *Youn and Price, 2009*) have documented how expectations of faculty have expanded from having to excel in either teaching, research or service, to having to demonstrate excellence in all three, research continues to be the most highly valued aspect of faculty work (*Acker and Webber, 2016*;

*Green and Baskind, 2007*; *Macfarlane, 2007*). Teaching is typically valued less than research, despite teaching duties often representing more than half of the workload (*Diamond and Adam, 1998*), and service activities come a distant third (*Fischman et al., 2018*; *Foos et al., 2004*).

Where, then, in this context of ever-expanding responsibilities and emphasis on research, does a commitment to the public come into the RPT process? This depends, of course, on which concept of public one focuses on and what dimensions are emphasized. In 2010 one of us (GEF) and two colleagues offered four basic dimensions of publicness that are used in discussions about what it means for universities to fulfill their public missions (*Fischman et al., 2010*). Perhaps the most frequently used dimension is that which refers directly to the concept of public patronage in the sense that public universities

\*For correspondence: juan@alperin.ca

in the United States and Canada belong to, and are administered by, federal, state or provincial agencies such as a state's appointed board of regents. A second dimension relates to the widespread notion that public universities should be as close as possible to free of cost, or the cost should not be a barrier to access through the use of financial assistance. A third dimension of publicness stems from the belief that universities should operate with the mission of addressing general social problems, promoting the common good, and emphasizing the social contributions of educational achievement beyond the individuals' benefit of access to higher education. Finally, the publicness of a university requires addressing the notion of accountability: to whom are higher education organizations accountable? Who represents the public interest in assessing the public effectiveness of an organization?

Notably, the work of faculty members intersects with all these dimensions: a good deal of research and development activities are supported with public money (i.e., public patronage), even at private institutions (*NSF, 2016*); faculty labor in the form of teaching, research and service is supposed to serve the common good and address social problems (i.e., public good), for which universities in the US and Canada receive a tax-exempt status; and, perhaps now more than ever, faculty need to demonstrate the value of their work (i.e., public accountability), and are therefore subject to more intense public scrutiny. Faculty work is also related in multiple ways to keeping the costs of access (at least at public universities) as low as possible (i.e., public access). Among other things, faculty work intersects with this economic dimension through their salaries (which are directly linked to maintaining low fees and tuition), their work as administrators, and through the expansion of their fund seeking actions (including fundraising activities not related to research grants). As universities struggle to define their own publicness, how do faculty effectively manage their careers in ways that support the various dimensions of the public mission of universities?

There appear to be organizational tensions between demands for demonstrating the public value of scholarship (i.e., public accountability) and the focus on "high prestige" or "high impact" publications by RPT committees. If publicness is interpreted as promoting public good, we might expect there to be calls for research outputs to take forms that are more ready for public consumption (not just more publicly available). Yet, determining the "prestige" of a publication venue is usually done at the discretion of evaluation committees (*King et al., 2006*; *Seipel, 2003*), through ranked lists or tiers supplied by academic institutions (*Malsch and Tessier, 2015*), or directly through impact factors and other citation metrics that measure use only within other scholarly works (*Adler et al., 2009*; *Walker et al., 2010*). These measures of prestige and impact reinforce the most commonly found publishing formats and venues (e.g., journal articles, books and conference presentations), which do not usually serve public needs in the way other forms do more directly (e.g., blog posts, podcasts, public outreach events). In the sense of public patronage, we might expect the emphasis on publications to move towards the use of open access (OA) models with the public gaining access to the work they are funding. OA has indeed grown, with around 50% of the most recent literature being freely available to the public (*Archambault, 2018*; *Archambault et al., 2014*; *Piwowar et al., 2018*). Yet, OA remains low on the priority lists of faculty (*Dallmeier-Tiessen et al., 2011*; *Gaines, 2015*; *Odell et al., 2017*), even when surveys indicate that many faculty believe open access to their published works is beneficial to their careers due to wider readership (*Dallmeier-Tiessen et al., 2011*; *Gaines, 2015*; *Odell et al., 2017*). It seems these faculty simultaneously hold the conflicting belief that traditional publishing is better for their careers overall because it is valued more in the RPT process (*Migheli and Ramello, 2014*; *Peekhaus and Proferes, 2015*; *Peekhaus and Proferes, 2016*; *Rodriguez, 2014*).

The debate about OA and of where to publish has been complemented with a growing interest for scientific measures beyond citations (so-called altmetrics; *Priem et al., 2010*). Some hope that these new metrics might serve as indicators of societal impact (*Bornmann, 2014*; *Bornmann, 2015*; *Robinson-Garcia et al., 2017b*; *Konkiel, 2016*). However, despite predictions that there would be a movement towards using non-citation metrics to assess the influence of research findings for RPT (*Darling et al., 2013*; *Piwowar, 2013*), there are

concerns, limitations and challenges in the use of these metrics that are hampering their uptake (*Gordon et al., 2015*; *Haustein et al., 2016*; *Howard, 2013*; *Lopez-Cozar et al., 2012*). Moreover, there is little evidence that mentions on social media are correlated with citations (see *Konkiel et al., 2016* for an overview) or that they can serve as indicators of public uptake (*Alperin et al., 2019a*; *Didegah et al., 2018*; *Robinson-Garcia et al., 2017a*).

It may be, however, that interest in developing and adopting these new metrics is not indicative of an interest in measuring the alignment between research and the public, but of growing calls for public accountability. A recent independent report commissioned by the Higher Education Funding Council for England (HEFCE) to assess the role of metrics in research assessment and management sees this as part of a "metric tide" that has been swelling in part because of "growing pressures for audit and evaluation of public spending on higher education and research" (p. 136, *Wilsdon et al., 2015*). Within the RPT process, there is little evidence for the inclusion of altmetrics within formal evaluation procedures (*Gruzd et al., 2011*; *Howard, 2013*), although this may be changing, as examples of altmetrics in faculty CVs have begun to emerge (cf., *Webster, 2018*). However, even in some documented cases where they were included (information science and medicine), department chairs did not value them towards promotion (*Aharony et al., 2017*; *Cameron et al., 2016*; *Fischman et al., 2018*). In contrast, there is evidence that institutions consider citation counts in their RPT process, which, by design, only measure uptake and use of the research by the academic community (*Dagenais Brown, 2014*; *Harley et al., 2010*; *Reinstein et al., 2011*).

Another attempt to address the public dimensions of faculty work beyond publication and dissemination formats is manifested through concerted efforts to engage communities in the research process itself. Such efforts can be seen in the growing body of work about such practices (cf. a bibliography of over 600 articles on Community Engaged Scholarship; *CES Partnership Resources, 2014*), and in the various statements, toolkits and standards for documenting and evaluating community engaged scholarship in faculty RPT guidelines, including the Carnegie Foundation's Elective Community Engagement Classification, the Association of Public Land-Grant Universities' Task Force on the New Engagement, the Research University Engaged Scholarship Toolkit developed by The Research University Civic Engagement Network (TRU-CEN), and a partnership of eight Canadian universities that developed Rewarding Community-Engaged Scholarship: Transforming University Policies and Practices. While many faculty have embraced such community engaged scholarship and related models, there is still little evidence these are valued across the academy. In particular, Harley et al. found that faculty who find ways to give back to the community and acknowledge the support of taxpayer funding, such as by participating in public education, generally receive recognition for these efforts regardless of institution type or field of study (*Harley et al., 2010*). However, these kinds of activities, while representing valid social contributions that can increase a university's accountability to the public, are often not recognized formally in the RPT process (*Goldstein and Bearman, 2011*).

Although previous work provides a sense of how the dimensions of publicness outlined here (public patronage, public access, public good and public accountability) intersect with the RPT process (*O'Meara, 2002*), more empirical work is needed to understand how publicness is incentivized in faculty careers (*O'Meara, 2014*; *O'Meara et al., 2015*). To this end we set out to collect documents, including collective agreements, faculty handbooks, guidelines and forms, that describe RPT requirements for faculty at a representative set of higher education institutions in the US and Canada. We collected these documents to analyze the degree to which various terms and concepts, in particular those that relate to research outputs and assessment, are mentioned in the RPT process, and discuss how the presence of these terms may relate to different concepts of publicness in higher education.

## Materials and methods

### Selection of sample

We began by creating a stratified random sample based on the 2015 edition of the Carnegie Classification of Institutions of Higher Education for US-based institutions, with an eye to have representation of institutions identified as: 1) doctoral universities (i.e., research-focused), which we refer to as R-type institutions; 2) master's colleges and universities, which we refer to as M-type institutions; and 3) baccalaureate colleges, which we refer to as B-type. Each of these categories is made up of multiple subcategories. R-type institutions are subdivided into those

with highest research activity, higher research activity and moderate research activity (R1, R2 and R3); the M-type institutions are subdivided into larger programs, medium programs and small programs (M1, M2 and M3); and the B-type institutions are subdivided into those that are arts and science focused and those from diverse fields (Bas and Bd). For Canadian-based institutions, we used the 2016 edition of the Maclean's Rankings, which similarly classifies institutions into: 1) doctoral (R-type); 2) comprehensive (M-type); and 3) undergraduate (B-type). We aimed to have enough institutions in each of the three broad categories to have statistical power of .8, assuming a small effect size (.25 of a standard deviation), when broken down by discipline. A summary of the number of institutions in each category, the number that we included in our random stratified sample, and the number for which we were able to obtain documents can be found in *Table 1*.

We collected documents that applied to the institution as a whole, and also those that applied to specific departments, schools or faculties, which we collectively refer to as academic units. We made a concerted effort to collect documents from academic units from a wide range of disciplines. While there is no single accepted classification system for fields of study, we opted to use the structure of fields and their subfields provided by the National Academies Taxonomy to group disciplines into three main areas: Life Sciences (LS); Physical Sciences and Mathematics (PSM); and Social Sciences and

Humanities (SSH; *National Academy of Sciences, 2006*).

## Collection of documents

We set out to collect documents from the institutions identified. In November 2016 we put out calls on social media and on several mailing lists related to issues of scholarly communications and librarianship, but when that method failed to yield many documents, we turned to a more proactive approach. Equipped with the randomly selected list of institutions, we searched the web for the documents. This method was especially fruitful for identifying documents about RPT that are set out by the institution, but not by individual academic units. For the latter, we searched for email addresses of faculty members of units at each of our target institutions by navigating from their university webpages to those of different faculties and their departments, making sure to look at departments from across the three fields. Given the variety of units, organization structures and naming conventions, our selection of which units to target was not perfectly systematic. It was impossible, for example, to target a specific unit by name across different institutions, since each university makes different decisions of whether to put a discipline within its own department, school or faculty (if it even has a unit to correspond with the discipline at all). Instead, we focused on the concept of an "academic unit" as any administrative unit within the university structure, and from those units listed on websites, our research assistant

**Table 1.** Sampling summary of universities from Canada and the United States.

| | | Number in category | Number sampled | Percent sampled | Number with documents |
|---|---|---|---|---|---|
| R-type | R1 | 115 | 17 | 15% | 15 |
| | R2 | 107 | 16 | 15% | 15 |
| | R3 | 113 | 17 | 15% | 14 |
| | RCan | 15 | 15 | 100% | 12 |
| M-type | M1 | 393 | 17 | 4% | 11 |
| | M2 | 298 | 12 | 4% | 10 |
| | M3 | 141 | 6 | 4% | 4 |
| | MCan | 15 | 15 | 100% | 13 |
| B-type | Bas | 259 | 14 | 5% | 11 |
| | Bd | 324 | 17 | 5% | 5 |
| | BCan | 19 | 19 | 100% | 17 |

Overview of population of universities from the United States and Canada by type and sub-type, the number and percent randomly chosen for the stratified sample, and the number of institutions for which at least one relevant document was obtained.

DOI: https://doi.org/10.7554/eLife.42254.002

attempted to pick contacts across the three main field categories.

In the end, we sent at least 915 emails to faculty from a dedicated project account between late 2016 and August 2017. In many cases, the persons contacted did not reply, the email address was no longer valid, or there was an auto-response. In many others, the faculty responded to let us know that they were not aware of any documents for their academic unit. In other instances, the person contacted responded with documents pertaining to their unit, and, in a few cases, with documents for several units at their institution.

As a result of this process over an almost year-long period, we obtained 864 documents from 129 universities and 381 units, of which 98 (25.7%) are from LS; 69 (18.1%) are from PSM; 187 (49.1%) are from SSH; and 27 (7.1%) are from multidisciplinary units that could not be classified under a single category. A large proportion of the documents collected are undated, but some have dates that go back as far as 2000 and as recent as the year of collection. To the best of our knowledge, these are the documents that the sender believed to be the most recent or applicable. While these documents correspond to the different types of universities and fields, the units are not spread out across all the universities evenly. We have at least one unit-level document from 60 of the 129 universities. We were told that documents did not exist at the academic unit-level by at least one faculty member at the remaining 69. In the majority of cases, we have four or fewer unit-level documents from each institution, but there are 10 instances in which we have more than 10 unit-level documents per institution (with a maximum of 45).

### Identification of terms

We proceeded to load these documents into QSR International's NVivo 12 qualitative data analysis software as two separate sets: the documents corresponding to university-level policies and those corresponding to different academic units. First, we created an NVivo "case" for each institution and academic unit, and we included in these "cases" the content of their respective documents. We then searched the documents for terms of interest, sometimes grouping several terms under a single concept, using various strategies as described in the research methodology notes found in the public dataset (*Alperin et al., 2018*). The mentions of each term or concept were included in an NVivo "node."

We subsequently performed a "matrix coding query" in NVivo to export a table with every university and academic unit as a row, and each of the nodes (terms and concepts of interest) as a column. Matrix coding queries show the intersections between two lists of items, with each cell in the matrix marked with whether at least one document from that university or academic unit had at least one mention of the corresponding term. Using this matrix, we were able to write a Python script to merge the data with the sample descriptors and calculate counts and percentages of universities and academic units that mentioned each term, and split those across university types and fields. We were also able to combine the results of the university and academic unit-level analyses to provide counts of whether a term was mentioned in at least one academic unit or one university-level document for each university. Unless otherwise specified, the results that follow report this combined analysis. The code used to generate these counts can be found in *Alperin, 2019b* (copy archived at https://github.com/elifesciences-publications/rpt-project).

For each term and concept, we used a chi-square analysis of contingency tables to determine whether the frequencies across categories were significantly different from a uniform distribution. For all analyses, the null hypothesis was that the overall proportion of documents containing the term or concept was the same between the different categories. In *Figures 1* and *6*, which compared institution types, the null hypothesis represented R-type=M-type=B-type. In *Figures 3* and *8*, which compared disciplines, the null hypothesis represented Social Sciences/Humanities=Physical Sciences/Math=Life Sciences=Interdisciplinary. The alternative hypothesis was that the proportion of documents containing the term/concept of interest was not equal across all different categories included in the test. Statistically significant differences are indicated in *Figures 1*, *3*, *6* and *8* with the following symbols: *: $p<0.05$; **: $p<0.01$; and ***: $p<0.001$. For all panels in *Figures 2* and *7*, and some portions of *Figures 1*, *3*, *6* and *8* (noted in the figure captions) the data did not meet the assumptions of chi-square analysis, namely an expected frequency of at least 5 for all conditions.

The data that support the findings of this study are available in the Harvard Dataverse with the identifier https://doi.org/10.7910/DVN/

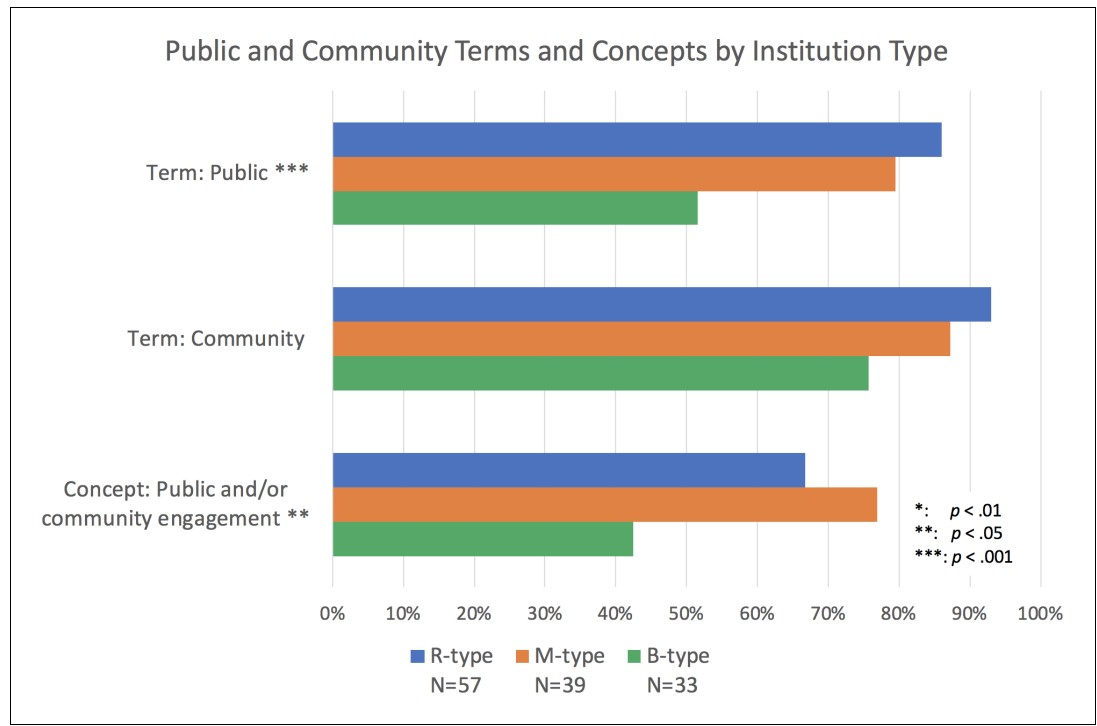

**Figure 1.** Percentage of institutions mentioning public and community terms and concepts by type of institution. Bars represent whether each term or concept (several terms and phrases) was identified within documents from doctoral/research-focused universities (R-type; blue), master's colleges and universities (M-type; orange) and baccalaureate colleges (B-type; green). The terms "public" and "community," and the concept of "public and/or community engagement" appear less often in documents from B-type institutions than from M- and R-type. The conditions of the chi-square test were not met for the term "community," but the chi-square analysis reveals the difference in presence of term "public" and concept "public and/or community engagement" are significant. Chi-square tests: Term Public: $\chi^2$ (2, $N$=129)=13.85, p<0.001; Concept Public and/or community engagement: $\chi^2$ (2, $N$=129)=9.61, p<0.01.
DOI: https://doi.org/10.7554/eLife.42254.003

VY4TJE (*Alperin et al., 2018*; *Alperin, 2019b*). These data include the list of institutions and academic units for which we have acquired documents along with an indicator of whether each term and concept studied was found in the documents for the institution or academic unit. The data also include the aggregated values and chi-square calculations reported. The code used for computing these aggregations can be found on Github (*Alperin, 2019b*). The documents collected are available on request from the corresponding author (JPA). These documents are not publicly available due to copyright restrictions.

## Results: public and community

We began our analysis with the terms "public" and "community" to understand the degree to which the public is talked about, and to gain a sense of the context surrounding their inclusion in RPT documents. We then focused on several

terms and groups of terms (i.e., concepts) that intersect with the notions of publicness identified above, starting with the concept of "public and community engagement," the presence of which would be indicative of incentives to work alongside the public in ways that more closely align research to the public's needs. Given the importance assigned to research in the RPT process (*Schimanski and Alperin, 2018*), we then turned our attention to terms and concepts related to research publications and their assessment, such as "open access," "publication formats" and "metrics," all of which speak to the different aspects of publicness outlined above.

### Context surrounding public and community

In analyzing RPT documents for their inclusion of concepts related to the public and community, we found that 87% of institutions mention the term "community" in either the university level

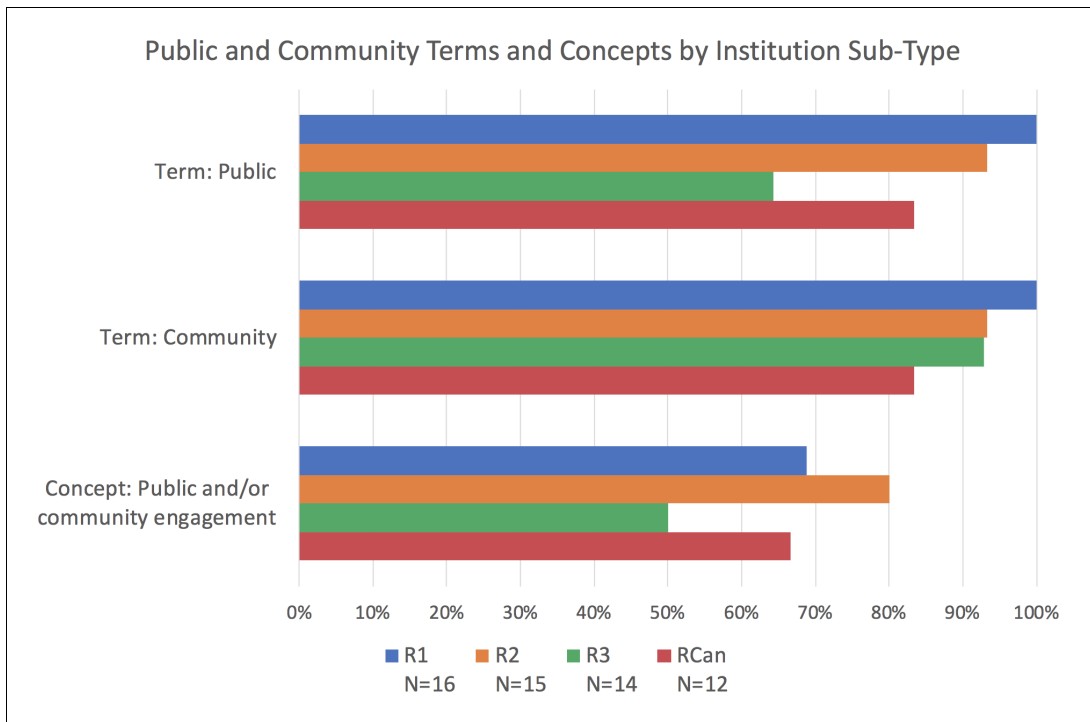

**Figure 2.** Percentage of institutions mentioning public and community terms and concepts by institution sub-type. Bars represent whether each term or concept (several terms and phrases) was identified within documents of doctoral/research-focused universities, from the most research intensive (R1; blue), to those that are less so (R2; orange, and R3; green), as well as the Canadian research universities (RCan; red). The terms "public" and the concept of "public and/or community engagement" appear more at R1 and R2 institutions than R3, with RCan universities falling in the middle. However, sample sizes violate conditions for a chi-square test to measure the significance of these differences.

DOI: https://doi.org/10.7554/eLife.42254.004

or academic unit guidelines, while 75% mention the term "public."

Overall, inclusion of the terms "public" and "community" is most common in research-focused (R-type) institutions (*Figure 1*). Within R-type institutions, we also found a trend towards greater inclusion of these terms at those institutions with the highest level of research activity (i.e., R1). All documents at the R1 level included the terms "public" and "community," while 93% of R2 institutions included both these terms and only 83% of the Canadian R-type (RCan; *Figure 2*). Within the academic units of R-type institutions, we found that of the disciplines examined, the Life Sciences (LS) most frequently include these terms, with 88% including "public" and 100% including "community" (*Figure 3*).

To better understand the context in which the terms "public" and "community" were being used, we analyzed the most frequent words surrounding each term. With these and other terms, we considered a word to be near our term of interest if it was within the 15 words preceding or following it (the length of an average sentence; *The Acropolitan, 2017*). The 10 most used words surrounding "public" were, in descending order of frequency: "service," "faculty," "professional," "research," "university," "activities," "teaching," "community," "work" and "academic" (*Figure 4*). The 10 most used words around "community" were, in descending order of frequency: "university," "service," "faculty," "professional," "academic," "research," "activities," "members," "teaching" and "member" (*Figure 5*).

The high incidence of these terms suggests that publicness features in the RPT process in some way. Although both "service" and "research" appear among the most frequent words surrounding "public" and "community," "service" is mentioned 1,170 times near "public" and 4,184 times near "community," while "research" is mentioned less than half as much (668 times near "public" and 1,671 near "community"). This, and the other frequent words in

 

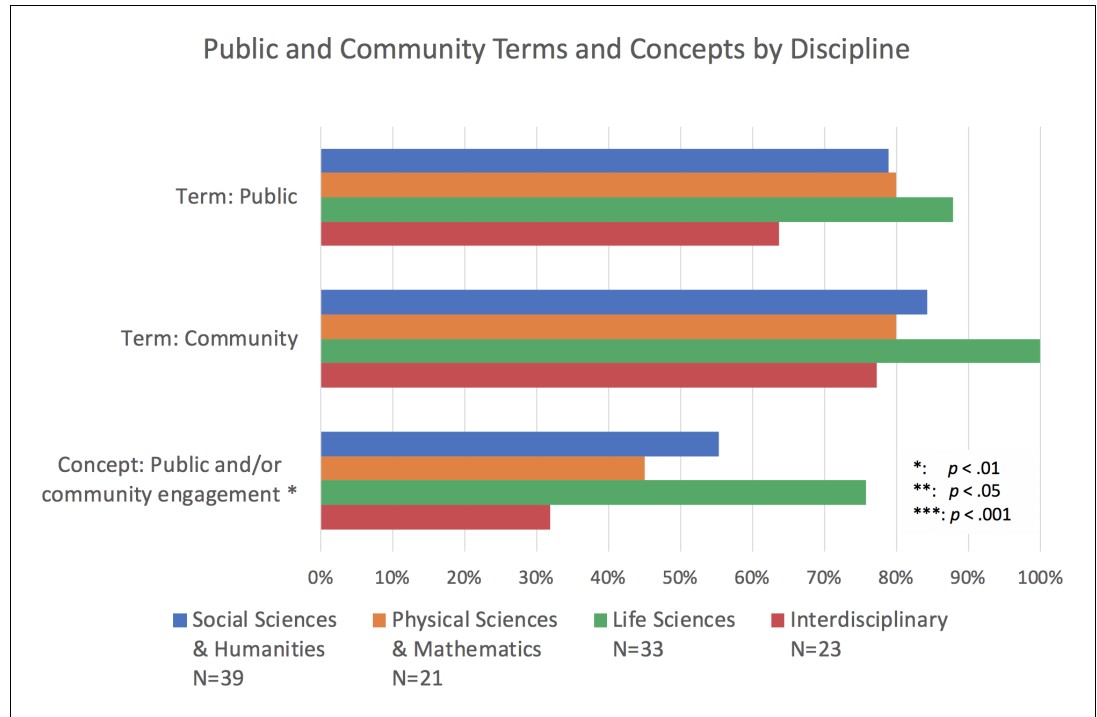

**Figure 3.** Percentage of institutions mentioning public and community terms and concepts by discipline. Bars represent whether each term or concept (several terms and phrases) was identified within documents of academic units from Social Sciences and Humanities (SSH; blue), Physical Sciences and Mathematics (PSM; orange), Life Sciences (LS; green) and multidisciplinary units (red). The terms and concepts appear more frequently in LS units than others. Sample size conditions for a chi-square test were only met for the concept of "public and/or community engagement," where it indicates that the difference in this category is significant. Chi-square test: Concept Public and/or community engagement: $\chi^2$ (3, $N$=116)=12.45, p<0.05.

DOI: https://doi.org/10.7554/eLife.42254.005

the surrounding context, are indicative of the terms "public" and "community" being more commonly associated with the service component of RPT, which is the least highly regarded of the RPT trifecta (*Fischman et al., 2018*; *Foos et al., 2004*; *Harley et al., 2010*).

Instances of "service" near the word "public" often included references to "public service" as a dimension or set of activities within the service category, thus explicitly separated from research. For example, guidelines of the Faculty of Arts of the University of Regina state that "The duties of a faculty member shall normally include: teaching and related duties (hereinafter "teaching"); scholarship, research, or equivalent professional duties (hereinafter "scholarship"); participation in collegial governance (hereinafter "administrative duties" and/or public service)" (*University of Regina, 2017*). Similarly, guidelines of the College of Education and Behavioral Sciences at the University of Northern Colorado state that "American colleges and universities have customarily examined faculty performance in the three areas of teaching, scholarship, and service, with service sometimes divided further into public service and service to the college or university" (*University of Northern Colorado, 2010*). While establishing this separation between (public) service and research, some documents also mandate the relatively lower importance of this and other dimensions of service in comparison to research activities. For example, the guidelines of the Department of Economics at the University of Utah manifest that "The Department's criteria that pertain to the qualification of candidates for retention, promotion, and tenure at all levels are: research, teaching, and university, professional, and public service. Research and teaching are of primary importance in evaluating the actual and potential performance of a candidate. Service is of secondary importance, but adequate performance in this area is expected of all candidates" (*University of Utah, 2007*).

In the case of "community," it becomes apparent by looking at its frequent proximity to

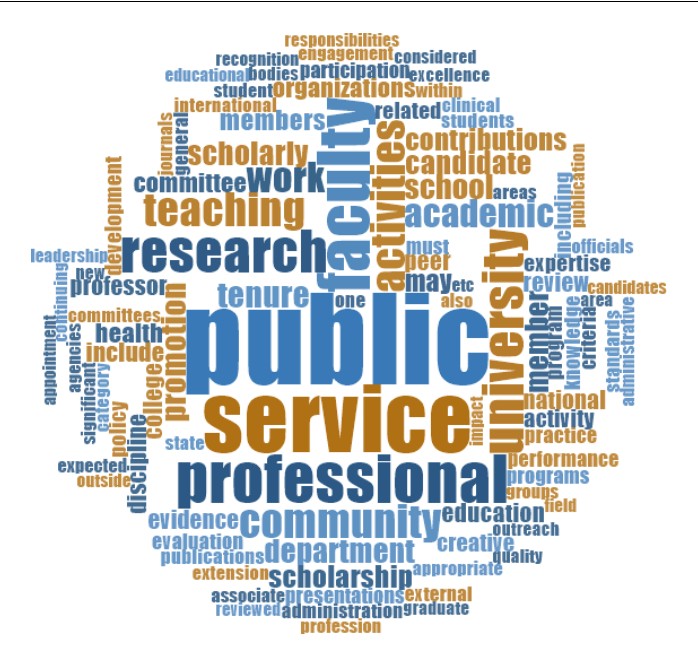

**Figure 4.** Relative frequency of words surrounding the term "public". Visual representation of the relative frequency of words near (within 15 words) the word "public" across all documents. The most frequent word near "public" is "service." Along with other frequent words, this suggests that in the context of RPT, "public" is most often associated with a service activity.

DOI: https://doi.org/10.7554/eLife.42254.006

words like "university," "service," "faculty," "professional" and "academic" that this term is generally used to refer to the academic community, composed primarily of faculty members (*Figure 5*). Again, we often see a requirement to provide service to this particular community in statements such as the following: "Distinctive service to the University and academic community would be evidenced by the candidate having made contributions of leadership and innovation involving decisions and policies that have had a major beneficial influence" (*Acadia University, 2014*); and "All candidates for tenure will be expected to demonstrate … that he/she has become a responsible and contributing member of the University/academic community" (*Simon Fraser University, 2013*).

Although these terms are also found within the context of research, as in some of the quotes above, we noted that the word "research" can appear near the words "public" and "community" without being directly relevant to the notion of public and/or community engaged research. This motivated a more refined coding strategy for this concept.

## Public and/or community engagement in research and scholarship

To better understand how engaging the public and the community in the research process is valued, we collected mentions of the concept of public and community engagement, using the variants identified by Barreno et al. as a foundation (*Barreno et al., 2013*). We collected references containing the following keywords: "community engagement," "scholarship of application," "scholarship of engagement," "community(-)engaged" [scholarship and research], "engaged scholarship," "engaged research," "community(-)based" [research, teaching and service], "community outreach," "applied scholarship," "public engagement," "public outreach," "public scholars," "public scholarship," "community scholarship" and "knowledge mobilization." We also conducted snowball searches based on derivations of the keywords – for example, after searching for "public engagement," we searched for variants such as "publicly engaged [scholarship, research]," "engaging the public," "engaging the community" and "engaging communities." In order to ensure that we were covering as many variants springing from the above keywords as possible, we also searched for instances in which the words "public" and "community" were found in proximity (three words distance) to "scholarship," "engagement," "research," "application" and stemmed words ("engaged," "engaging," "researching," "applied," "applying") and conducted manual revision and coding of relevant references. Furthermore, we also revised and manually coded every mention of "public" and "community" to identify more general instances in which the idea of engaging the public and/or the community was present.

To encompass all the phrases coded by this strategy, we chose to use the term "public and/or community engagement in research and scholarship." This term is intended to capture mentions that are more specific than the individual terms "public" and "community" while being more inclusive than the widely accepted definitions of "community engaged scholarship" found in the literature and in places like the Carnegie Community Engagement Classification. We found 64% of institutions in our sample include at least one mention of this expanded concept of public and/or community engagement within their RPT documents, most

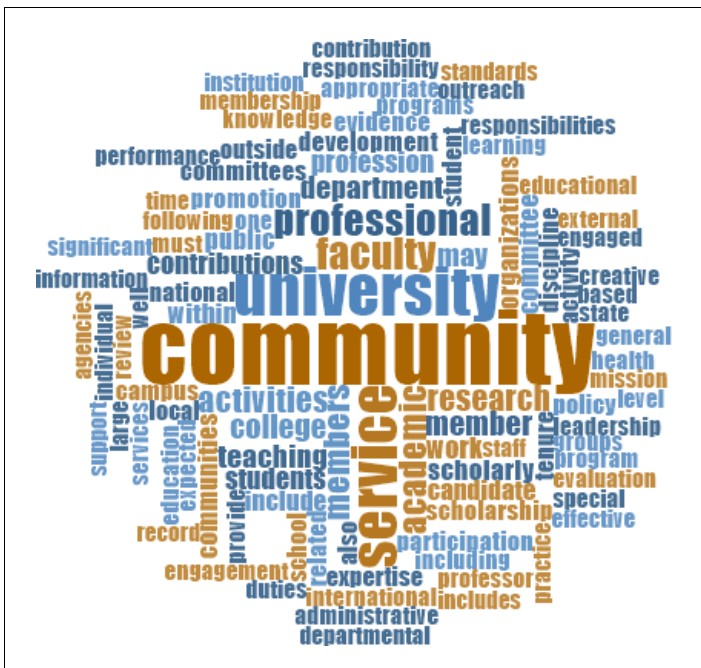

**Figure 5.** Relative frequency of words surrounding the term "community". Visual representation of the relative frequency of words near (within 15 words) the word "community" across all documents. The most frequent word near "community" is "university." Along with other frequent words, this suggests that the community most often referred to is that of other academics.

DOI: https://doi.org/10.7554/eLife.42254.007

commonly in the master's colleges and universities (M-type) institutions (*Figure 1*).

Within R-type institutions, the concept of public and/or community engagement was more common in the documents of R2 type institutions than in those of R1, R3 and RCan subtypes. Like with the terms "public" and "community," the concept of public and/or community engagement was most common in the LS (at 76% of those academic units; *Figure 3*).

In some academic units this work is still seen as a service-related activity. For example, guidelines of the Faculty Division of Biological Sciences of the University of Wisconsin-Madison classify the academic activity required of the candidate in "teaching, research, and outreach including extension, community engaged scholarship and service" (*University of Wisconsin-Madison, 2016*). However, community and/or public engagement is often considered a component of research and scholarly activities. For example, guidelines of the Department of Political Science at the University of Guelph state "Community engaged scholarship involves mutually beneficial partnerships with the community (community may be defined as the local

community, but it may also be communities of interest that are local, national, or international in scope) that results in the creation of scholarly products. It is "engaged" in the sense that it involves forming campus-community collaborations in order to conduct scholarly research, evaluate social impacts and mobilize knowledge to address and solve problems and issues facing communities" (*University of Guelph, 2012a*). This particular instance, and others like it, draw their definition of community engaged scholarship from the Carnegie classification described earlier, which explicitly asks if community engaged scholarship is rewarded in faculty promotion. In our sample, 85 institutions (all from the United States) had opted to have their community engagement assessed, and 34 had attained the classification.

Similarly, guidelines from Thomas University say "The Scholarship of Application encompasses scholarly activities that seek to relate the knowledge in one's field to the affairs of society. Such scholarship moves toward engagement with the community beyond academia in a variety of ways, such as by using social problems as the agenda for scholarly investigation, drawing upon existing knowledge for the purpose of crafting solutions to social problems, or making information or ideas accessible to the public" (*Thomas University, 2016*). This last quote shows how community engaged scholarship is expected to orient research activities towards serving the public good while explicitly requesting that the ideas developed as a result of the research become publicly accessible.

If public and/or community engaged scholarship activities are strongly linked to notions of publicness, it is important to understand to what degree these activities are valued in the RPT process. Although some institutions consider this work as valuable as "traditional research," others do not regard it as relevant. On the one hand, documents like that of the University of Windsor declare that "Research and Scholarly activities may include traditional research with traditional dissemination venues and publicly engaged academic work that creates knowledge about, for, and with diverse publics and communities with traditional and non-traditional dissemination venues" (*Windsor University, 2016*). On the other hand, guidelines of the Faculty Division of Physical Sciences at the University of Wisconsin-Madison established enhancing "public engagement in the physical sciences" among "professional service" activities, but go on to specify that "significant contributions in the form

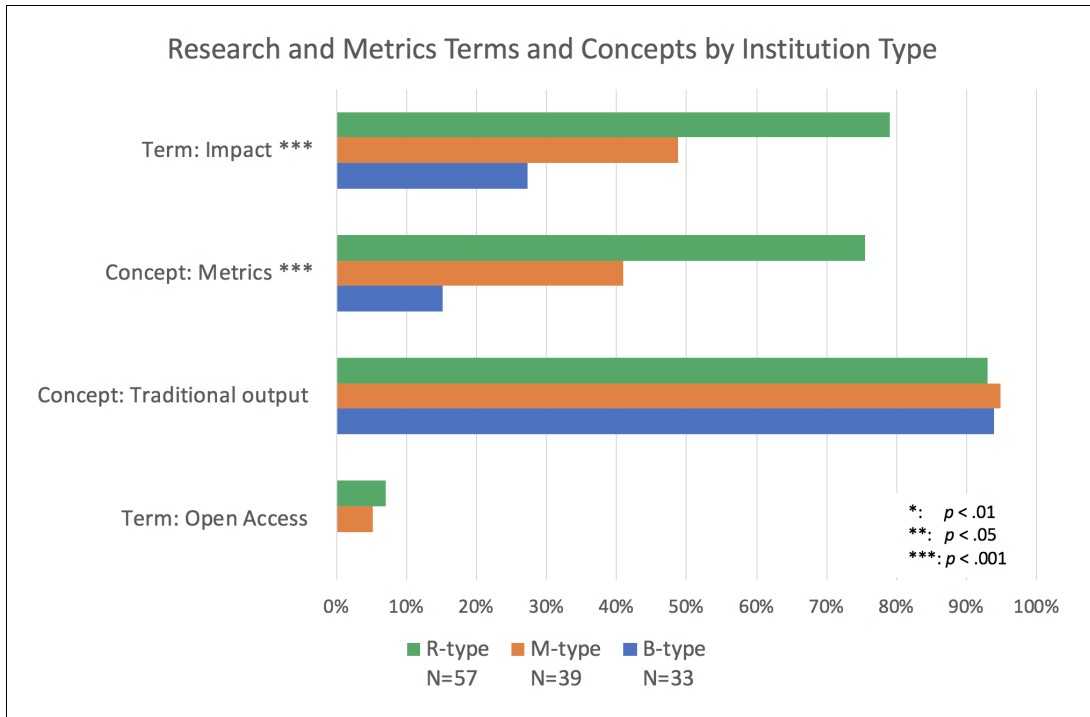

**Figure 6.** Percentage of institutions mentioning terms and concepts related to research and metrics by institution type. Bars represent whether each term or concept (several terms and phrases) was identified within documents from doctoral/research-focused universities (R-type; blue), master's colleges and universities (M-type; orange), and baccalaureate colleges (B-type; green). Chi-square analysis suggests that the term "impact" and the concept of "metrics" is more present at R-type than at M-type, and more present at M-type than B-type. The concept of "traditional outputs" is present at over 90% of each type, although the conditions for a chi-square test were not met for this concept or for the term "open access." Chi-square tests: Term Impact: $\chi^2$ (2, $N$=129)=24.13, p<0.001; Concept Metrics: $\chi^2$ (2, $N$=129)=32.04, p<0.001.

DOI: https://doi.org/10.7554/eLife.42254.008

of professional service can strengthen but may not serve as the basis for the candidate's case" (*University of Wisconsin-Madison, 2014*).

We start to see in such texts an explicit elaboration of the differences between public service and applied research or scholarship, with guidelines like those of Kalamazoo College drawing a clear distinction between the two by stating, "While most scholarship of engagement could also be considered public service, most public service is not scholarship of engagement. To be viewed as scholarship, the work must flow directly out of one's (inter)disciplinary expertise and involve the generation of new ways of thinking" (*Kalamazoo College, 2016*). Similarly, guidelines of the Department of Geography and Geology at the University of Southern Mississippi state: "The basic problem centers on the interpretation of applied research versus service … The Department defines applied research as the movement of new or innovative knowledge from the research community to the practitioner

community … Applied research may include both funded and non-funded efforts which result in the preparation and distribution of a manuscript or map; the publication of a professional paper, especially a peer-reviewed publication, book monograph or volume; the presentation of a paper before a professional organization; or the publication of a document submitted to a funding agency through grant or contract, where the document has been subjected to rigorous review and approval, and exhibits new and/or innovative approaches to the solving of a problem or the reporting of an outcome learned from lengthy and rigorous scholarly investigation" (*University of Southern Mississippi, 2010*).

## Results: research and metrics

While the context surrounding the concepts of public and/or community engaged scholarship allows us to see some of the ways in which faculty are asked to align their activities with the

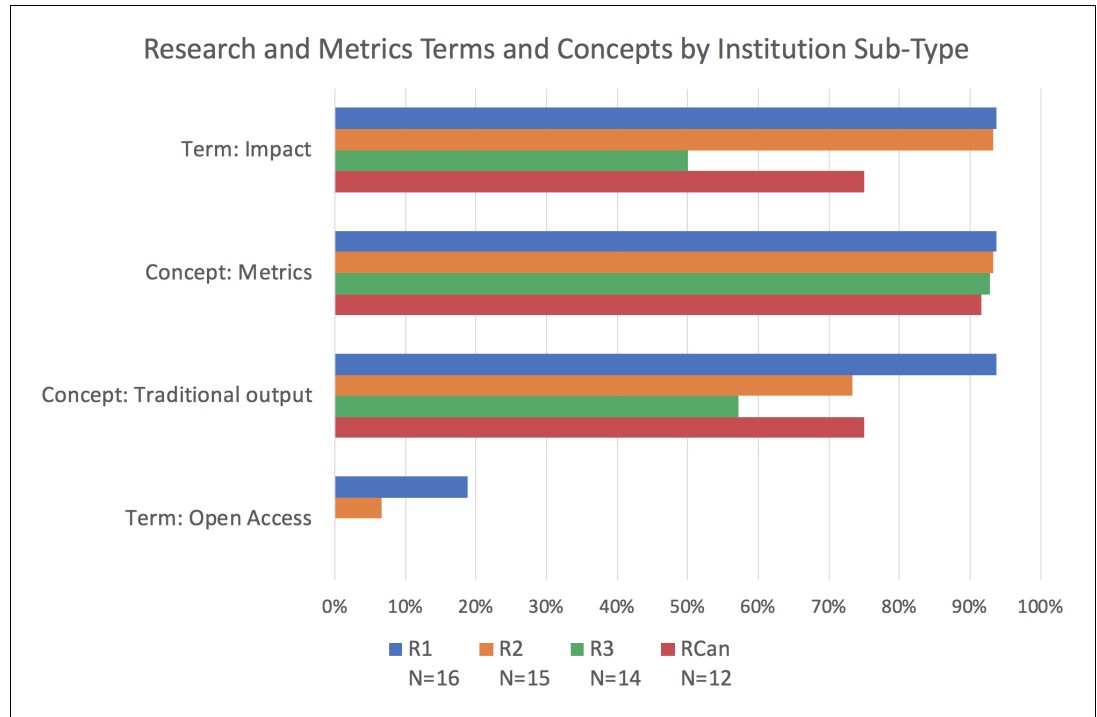

**Figure 7.** Percentage of institutions mentioning terms and concepts related to research and metrics by institution sub-type. Bars represent whether each term or concept (several terms and phrases) was identified within documents of doctoral/research-focused universities, from the most research intensive (R1; blue), to those that are less so (R2; orange, and R3; green), as well as the Canadian research universities (RCan; red). The term "impact" appears less in R3 institutions, and the concept of "metrics" appears to decrease with research intensity (with RCan institutions at similar levels to the R2 institutions from the US) However, the conditions for a chi-square test were not met to measure the significance of these differences.

DOI: https://doi.org/10.7554/eLife.42254.009

public good, the demarcation between this form of scholarship and "traditional research" suggests that we need to look at how the latter is discussed in the RPT guidelines separately. We therefore searched for mentions of traditional research outputs (which, as indicated above, are not typically geared towards being accessed and engaged by diverse audiences without specialized training), and whether these outputs are expected to be made publicly available (through open access), what type of impact this work is expected to have (public or otherwise), and how it is evaluated. To do this, we conducted a similar analysis to that above, but with terms related to traditional research outputs, open access, impact and citation metrics, and considered their prevalence in relation to public and community terms.

### Traditional research outputs

We found that guidelines for faculty often provide specifics when it comes to the types of research outputs that can be considered for tenure and promotion. This frequently takes the form of a list of outputs that are considered valuable, although these lists sometimes also explicitly mention that other forms of scholarship are welcome. For instance, guidelines of the College of Business and Economics at Boise State University manifest that "Examples of the types of evidence which demonstrate research and scholarly activity include (but are not limited to): (1) Articles in refereed journals (2) Books or research monographs (3) Chapters in books or monographs (4) Other published articles (5) Papers presented at academic conferences and/or published in proceedings (6) Published book reviews (7) Participation as a paper discussant or panel discussant at academic conferences (8) Grants and contracts for research and scholarly activities" (*Boise State University, 2008*). Similarly, guidelines of Memorial University of Newfoundland establish that "Factors that may be considered [as a demonstrated record of research, scholarship, or creative and professional activities] include but are not limited to:

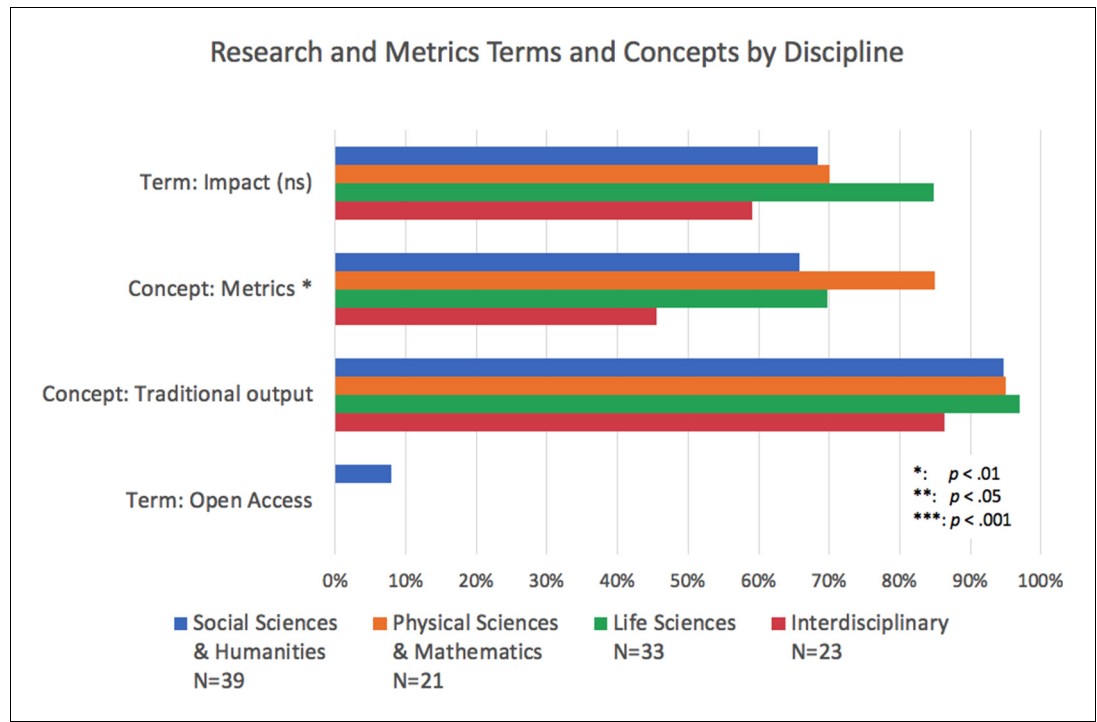

**Figure 8.** Percentage of institutions mentioning terms and concepts related to research and metrics by discipline. Bars represent whether each term or concept (several terms and phrases) was identified within documents of academic units from Social Sciences and Humanities (SSH; blue), Physical Sciences and Mathematics (PSM; orange), Life Sciences (LS; green) and multidisciplinary units (red). The concept of traditional outputs is present in the vast majority of units. The term impact is more present in LS, but a chi-square test suggests the difference is not significant. The chi-square analysis also indicates the difference in the presence of the concept of "metrics" (with PSM units mentioning it the most) is significant. The conditions for a chi-square test were not met for other terms and concepts. Chi-square tests: Term Impact: $\chi^2$ (3, $N$=116)=5.75, p>0.05 (not significant). Concept: Metrics: $\chi^2$ (3, $N$=116)=7.33, p<0.05.
DOI: https://doi.org/10.7554/eLife.42254.010

the publication of books, monographs, and contributions to edited books; papers in both refereed and non-refereed journals; scholarly presentations delivered at professional meetings; success in grant competitions; participation in panels; unpublished research including current work in progress both supported and non-supported; editorial and refereeing duties; creative works and performances; and scholarship evidenced by the candidate's depth and breadth of knowledge and general contributions to the research life and creative milieu of the University" (**Memorial University of Newfoundland, 2014**).

When looking for traditional outputs (i.e., books, conference proceedings, grants, journal articles, monographs and presentations), we found at least one mentioned in 90–95% of R-, M- and B-type institutions, in all R-sub-types, and in the three disciplinary categories (it was a little below 90% for the interdisciplinary

academic units; **Figures 6–8**). Of the terms analyzed in our study, this group related to outputs was the most consistently present across institution types and disciplines. Their consistency and relative ubiquity show that if there is one thing that is certain to count towards faculty career progression, it is producing traditional academic outputs.

Meanwhile, other outputs resulting from faculty work that relate to the public and/or the community are sometimes considered as a service activity. For example, traditional outputs and metrics are mentioned in the "Scholarship and Research" section of the Institute of Environmental Sustainability at Loyola University's Tenure and Promotion Guidelines, while "publishing articles for the general public" is included within the section "Professional Contributions" (which are deemed as "all service and accomplishments not defined as research … and can contribute to the general development of the

broader profession"; *Loyola University, 2015*). Similarly, guidelines of the Department of Psychology of the University of Guelph establish that "Normally, publication of scholarly works relevant to some aspect of the discipline of psychology will be considered. Other publications (e.g., trade books, articles in popular magazines) will be evaluated under service to the community. Where appropriate, however, these products may be referenced as knowledge mobilization activities in the dossiers related to scholarship, service or community engagement" (*University of Guelph, 2012b*). Like these, we found many instances where faculty are offered lists of valued outputs beyond those used for communicating within the academic community, but more often than not, these are not regarded as research activities.

### Open access

Since traditional outputs are the ones most valued, and since these outputs are not typically geared towards the public, we searched for evidence that universities sought to at least grant the public access to these scholarly works. Although the number of articles freely available to the public has been growing from year to year (*Archambault, 2018*; *Archambault et al., 2014*; *Piwowar et al., 2018*), we found only a handful of mentions of "open access" across the hundreds of documents we studied.

Only 5% of institutions explicitly mentioned the term in their guidelines, with most of those mentions (4 of 6) in R-type institutions and the rest (2 of 6) in M-types (*Figure 6*). Open access was not mentioned at all in B-type institutions. Notably, of those mentions that occurred within academic unit documents (as opposed to those that apply to the institution as a whole), all three of them were in SSH units (*Figure 8*).

Contrary to our expectation that these mentions would promote public access to research outputs, we found the majority of these few instances call for caution around publishing in OA venues. This caution appears to stem from a focus on, or misunderstanding of, OA an as inherently predatory publishing practice (OA refers to free and unrestricted access to and re-use of articles, not to a business model; *BOAI, 2002*) and assumes that OA journals do not utilize peer review (even though 98% of the over 12,000 journals in the Directory of Open Access Journals perform some form of peer review). For example, the Department of Political Sciences at the University of Southern Mississippi notes that "Faculty are strongly cautioned against publishing in journals that are widely considered to be predatory open access journals" (*University of Southern Mississippi, 2016b*). The faculty handbook at the same university also explicitly calls out the practice of "using online journals which feature "instant publishing" of articles of questionable quality for a fee... described as "predatory open-access journals" for padding portfolios that received a negative evaluation (*University of Southern Mississippi, 2016a*). Similarly, the Department of Anthropology at Purdue University also associates open access publications with a lack of peer review by stating that "self-published, inadequately refereed, open-access writing, or online publications will be scrutinized carefully, and may be given little or no stature as evidence of scholarly accomplishment" (*Purdue University, 2014*).

Other universities and academic units use less negative language, while still calling for caution around OA. Across several instances, it is strongly implied that it is a rigorous peer review process that confers value to an OA publication, not the increased access that it grants to the public. The Department of Sociology at the University of Central Florida is the most explicit in this regard stating that "some of them [open access journals] are peer-reviewed and of very high quality, and some of them are not. The critical issue for tenure and promotion is neither the medium of publication nor the business model of the publisher but the rigor of the peer review process and the quality of the papers" (*University of Central Florida, 2015*).

It is also notable that none of the mentions of OA actively encourage or explicitly value open access. The closest that a document comes to encouraging open access is the Report of the UNC Task Force on Future Promotion and Tenure Policies and Practices from the University of North Carolina at Chapel Hill, which includes a link to a website from the UNC-CH Health Sciences Library that promotes OA (*University of North Carolina at Chapel Hill, 2009*). Beyond that, the most positive message faculty are receiving about OA – in the very few places where they are receiving any message at all – is that open access publications "may be meritorious and impactful" (*San Diego State University, 2016*), and that "Open-access, peer-reviewed publications are valued like all other peer-reviewed publications" (*University of Central Florida, 2014*).

*Impact*

We went on to examine the kind of impact that is expected of faculty in the RPT process to see if, despite the encouragement of traditional research outputs and the cautionary tone around open access, faculty are asked to have impact that goes beyond the academic community. We found that "impact" is a term of interest, with 57% of institutional documents mentioning it explicitly. Use of this term is most common in RPT documents of R-type institutions (79%; *Figure 6*) and, similar to "public" and "community," appears most frequently within higher-ranking R-type institutions (94% at R1, 93% at R2, and 50% of R3; RCan institutions fall in the middle at 75%; *Figure 7*). Related, we find similar results to "public" and "community" in that "impact" is mentioned most frequently within the documents of Life Sciences academic units of R-type institutions (85%; *Figure 8*).

Like with the other terms of interest, we assessed the most frequently employed words surrounding the term "impact" (within 15 words) in the RPT documents. The top ten are, in descending order of frequency: "research," "candidate," "work," "faculty," "quality," "teaching," "evidence," "field," "service" and "scholarly" (*Figure 9*). The term "public" is the 88th most frequent word near "impact," while "factors" and "factor" (likely referring to the Impact Factor) rank 67th and 204th respectively (discussed further below in the analysis of metrics).

We find a higher presence of "impact" in proximity to research related terms as compared to other RPT components. Although the associated words show that the impact of faculty work is a concern across all three areas of academic activity (research, teaching and service), "impact" is mentioned 904 times near "research" versus 392 times near "teaching" and 344 times near "service." It should be said, however, that how "impact" is defined is not always entirely clear, with several instances using non-specific descriptors, such as "major impact," "substantial impact," "demonstrable impact," "considerable impact," "significant impact," "valuable impact," "outstanding impact," "total impact," "maximum impact," "minimal impact" and various others. For example, guidelines of the University of Washington-Tacoma state that "Appointment with the title of professor of practice is made to a person who is a distinguished practitioner or distinguished academician, and who has had a major impact on a field important

to the University's teaching, research, and/or service mission" (*University of Washington-Tacoma, 2017*). Similarly, guidelines of the Department of Biological Sciences at Simon Fraser University manifest that "The number of publications is important, but secondary to their quality and total impact, and to the applicant's contribution to the research publications" (*Simon Fraser University, 2017*).

Meanwhile, the public dimension of impact, in any form, is minimally addressed. Specific mentions of this concept are rare (appearing in only 9% of the R-type institutions and 11% of the M-type), and are often non-specific about how that public impact will be determined. For example, guidelines of Carleton University establish that "Evidence appropriate to the discipline or field used to demonstrate the originality and quality of research/scholarly activity or creative work in support of an application for tenure or promotion may include, but is not limited to ... other publications demonstrating a high quality of scholarship with significant public impact" (*Carleton University, 2014*). Similarly, guidelines of the Faculty Division of Physical Sciences at the University of Wisconsin-Madison require faculty to "List the implications of the program; its relevance to the problems of agriculture, industry, and other segments of society in the state and nation; and its potential or demonstrated impact on the public" (*University of Wisconsin-Madison, 2014*).

*Metrics*

Since metrics are often cited as a common way to measure impact (*Reinstein et al., 2011*), we further analyzed the frequency of mentions of terms related to metrics, such as "citations,'" "impact factor," "acceptance/rejection rates" and the word "metrics" itself (see methodology note on *Alperin et al., 2019a*; *Alperin, 2019b* for the list of terms included). We found that 50% of institutions mention the concept of metrics at either the university level or the academic unit level. The mention of metrics within RPT documents is most common at R-type institutions (75%) as compared to M-type (41%) and B-type (15%; *Figure 6*). Within R-type institutions, mentions of metrics are more common at the higher-ranking institutions, with 94% of the documents of R1 institutions containing the concept, while only 73% and 57% of the R2 and R3 institutions contained the term (*Figure 7*). Again, Canadian R-type institutions fall in the middle of the range with 75% of those institutions mentioning the concept in their documents. Within

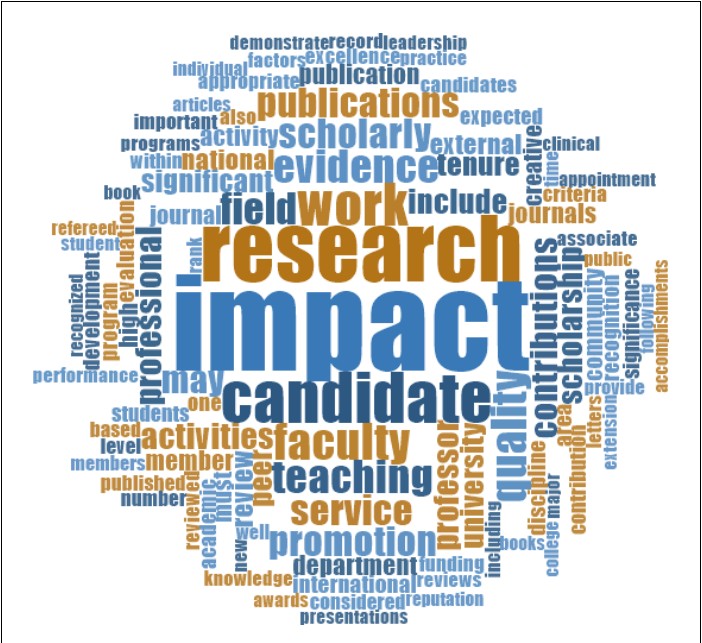

**Figure 9.** Relative frequency of words surrounding the term "impact". Visual representation of the relative frequency of words near (within 15 words) the word "impact" across all documents. The most frequent word near "impact" is "research." Along with other frequent words, this suggests that the type of impact most valued is that which relates to research activities.

DOI: https://doi.org/10.7554/eLife.42254.011

academic units of the R-types, we found greater mention of the concept in the documents of PSM units (85%) and the LS (70%) than in those of the SSH (66%; *Figure 8*).

The high incidence of terms related to citation metrics is indicative of the importance of measuring the use of the scholarly work by other scholars. We often found such terms associated with the notions of quality and impact, as in the case of the Department of Anthropology at the University of Utah that states "The candidate's research should be of high quality, showing originality, depth, and impact. In order to evaluate research quality, the departmental RPT committee shall evaluate the following: (. . .) number of citations per year in the Social Science Citation Index (SSCI) and, as appropriate, the Science Citation Index (SCI) . . . Candidates for tenure and promotion are expected to be visible in the citation indices, and their work should show evidence of continued impact" (*University of Utah, 2000*). Similarly, guidelines of Georgetown University include "Citation of a candidate's work in the professional literature, or other measures of scholarly impact" as indicators of "scholarly standing" (*Georgetown University, 2017*).

However, not all mentions of metrics endorse their use. For example, guidelines from the Faculty of Veterinary Medicine at the University of Calgary explicitly state that "Impact factors of journals should not be used as the sole or deciding criteria in assessing quality," and that "the reputation and impact of the journal . . . [should take] secondary consideration to the quality of the publication and the nature of the contributions" (*University of Calgary, 2008*). Other guidelines express how such measures are perceived, like the case of UC San Diego that "welcomes data on journal acceptance rates and impact factors, citation rates and H-index" while acknowledging that "some CAP [the Committee on Academic Personnel] members (as do senior staff of scholarly societies) retain various degrees of skepticism about such measures" (*University of California San Diego, 2015*). Yet, in some places where the guidelines recognize the "shortcomings of citation indices as measures of research impact," they continue to assert that "these remain important metrics within particular disciplines" (*McGill University, 2016*).

Only in rare cases do we find guidelines proposing the development of new metrics to evaluate publicly engaged academic work. Here, the University of North Carolina at Chapel Hill School of Information and Library Science stands out in stating that "Faculty are encouraged to present evidence of public engagement as part of their record and to suggest metrics or guidelines for assessing the impact and significance of the engagement" (*University of North Carolina at Chapel Hill, 2015*). This statement is supported by the Report of the UNC Task Force on Future Promotion and Tenure Policies and Practices that asks several questions, including "How public work must be count to [sic] as scholarship?" and concludes that "Answers to such questions have to be developed as departments and units create metrics by which to evaluate this work" (*University of North Carolina at Chapel Hill, 2009*).

This last example – the one that most directly discusses the relationship between metrics and the notion of public scholarship – highlights a conflict between two dimensions of publicness. On the one hand, public engagement and serving the public good are explicitly recognized and valued while, at the same time, the emphasis on metrics demonstrates how faculty are beholden to an accountability culture that relies predominantly on measurable and quantifiable outcomes. That metrics are used in this way is perhaps unsurprising, but their mention at three

quarters of R-type institutions is indicative of just how common the call is for citation measures as evidence.

## Discussion

Our research shows that, while there is a relatively high incidence of the terms "public" and "community" in the representative set of RPT documents – which could be interpreted as an indicator that faculty do need to consider the nature of the publicness of their work – there are neither explicit incentives, nor clear structures of support for assessing the contributions of scholarship to the various dimensions of publicness. Conversely, the higher incidence of mentions of traditional research outputs (which are not typically easy for the public to access and require specialized knowledge to be understood), the almost non-existent mentions of open access (which would grant the public access to all research), and the persistent presence of traditional citation metrics (which do not account for public use of scholarly work) indicate that, in order to be successful, faculty are mostly incentivized towards research activities that can be counted and assessed within established academic conventions.

Moreover, our analysis found that RPT documents signal that faculty should focus on uptake within their specific academic fields, especially at the R-type institutions where quantifiable citation-based metrics are mentioned in the documents of nearly three quarters of the institutions studied. As faculty careers are more closely scrutinized through metrics that seek to reflect research use and value within academia (i.e., citations; *Dahler-Larsen (2011)*; *Fischman et al., 2018*; *Hicks et al., 2015*; *Wilsdon et al., 2015*), the ability for faculty to dedicate time and energy into activities that more directly serve the public good are not incentivized.

We want to be very explicit that we do not oppose the use of well-defined indicators or metrics as one way (among many others) to assess the scholarly relevance of research. However, we suggest care is needed in identifying, and replacing, any simplistic policies that only pay lip service and symbolic attention to the public dimensions of scholarship and that inadvertently generate barriers to publicness by encouraging the use of poorly constructed metrics to assess research productivity. We are not the first to identify this need. The Humane Metrics Initiative (HuMetricsHSS; https://

humetricshss.org/about/), for example, has been working towards identifying metrics that support specific values, including engaging with one's community of practice and with the public at large. More broadly, there are many universities and individuals working on overcoming the limitations and the adverse effects that the use of metrics is producing; among these efforts, the Declaration on Research Assessment (*DORA, 2018b*) stands out with over 13,000 scholars and over 1,000 scholarly organizations as signatories who have expressed their commitment to avoiding simplistic models to assess scholarly impact (see also *Hicks et al., 2015*; *O'Neill, 2016*; *Simons, 2008*; *Vanclay, 2012*).

To this end, our work informs these efforts by identifying the specific modes of scholarship and assessment measures that are prevalent in current policies. We believe that our findings can help faculty reflect on how they focus their energies and characterize their efforts when they are being evaluated, while at the same time giving those conducting the evaluations (i.e., RPT committees, department chairs and deans) a greater understanding of how the guidelines at their institution may be inadvertently promoting certain forms of scholarship and assessment measures over others, which may be at odds with the public missions of many institutions. We suggest that, given the prominence of public and related terms in RPT documents and the lack of explicit metrics or incentives to encourage publicly-oriented scholarship, there are clear opportunities for institutions to reconcile these discrepancies.

Where we do find evidence for the promotion of specific forms of scholarship is in the types of outputs that are mentioned in the documents. While in this study we did not analyze all the outputs being asked of faculty, we found an almost ubiquitous presence of traditional research formats (i.e., books, conference proceedings, grants, journal articles, monographs and presentations) which are often not accessible to the public who ultimately underwrites the work. The remarkably consistent presence of these few terms across institution type and sub-type, and across disciplines, is likely not surprising to most readers, but is nonetheless a reminder of how entrenched these modes of scholarship are in academia.

What might be more surprising is the lack of positive mentions of OA as a way of facilitating the uptake of these deeply entrenched formats by a more diverse set of users through increased access. OA could be a bridge that links research activities, published in traditional formats, to

expanded engagement with more diverse groups of users and stakeholders, fulfilling the public patronage imperative of universities, but it does not advance a university's efforts along the other dimensions of publicness.

Our work thus highlights some of the ways institutions could better align metrics and incentives with the different dimensions of publicness to ensure these are adequately supported in the RPT process. It seems natural for those wanting to see change in the way public and/or community engaged scholarship is valued to want to see those changes reflected in the guidelines. Our findings show that these forms of scholarship are not always regarded as highly as "published research in top-ranked/High Impact Factor journals" and are often considered part of faculty service – the least valued aspect of the RPT trifecta. The lack of value placed on service creates disparity between faculty, something that warrants special consideration given that women spend more time on such roles, often at the expense of their career progression (*Guarino and Borden, 2017*; *Misra et al., 2011*).

Of course, there are different degrees of value placed on public and community engagement activities across universities and units. The appearance of terms related to public and/or community engaged scholarship in many of the guidelines from academic units from the Life Sciences (where medical schools are found), suggests that these forms of scholarship receive consideration in some of the fields where there are direct and obvious implications for the community, and where efforts are being made for more comprehensive models of research assessment (*Cabrera et al., 2018*; *Cabrera et al., 2017*).

Counting public and/or community engaged scholarship wholly as a research activity is just one way in which publicness could be better supported, but, as O'Meara states, "just because a college changes its written definition of scholarship in promotion policies does not mean that institutional members wake up the next day with a new view of faculty work" (p. 58, *O'Meara, 2002*). Other efforts, such as the requirement of the International Development Research Centre (IDRC) and other Canadian agencies for knowledge mobilization plans in all of their grants, are trying to promote publicness through funding incentives (*Lebel and McLean, 2018*). Others still have suggested expanding the RPT trifecta to introduce a new category that includes activities that aim to disseminate

information to a broader public, and that might be seen as a midpoint between research and service (*Harley et al., 2010*; *Scheinfeldt, 2008*). However, there may be limits to what additional categories related to "publicness" can achieve – particularly if they are not based on well-defined metrics – without understanding the limitations of the current assessment practices. Such categories may end up being undervalued in much the same way service is today.

Instead, our research confirms a discrepancy between how faculty work is assessed and incentivized through the RPT process and the stated goals of institutions to achieve scholarship for the public good. However, previous efforts at RPT reform suggest that solely changing what is written in RPT documents may not be sufficient to better align assessment practices and institutional goals (*O'Meara, 2005*). To close this gap, publicly orientated faculty work may first need to be considered on par with activities for which there are "quantifiable research metrics," since these are the ones that appear to be the most valued. That is, it seems difficult for faculty to carry out scholarly work aligned with the public dimensions of universities if this work is an additional burden that is separate from the main activity of producing knowledge.

Moreover, to close the gap, faculty may need to be allowed and likely encouraged to produce other types of outputs beyond the six traditional outputs we searched for. Relatedly, for the public availability of these and other outputs to be valued, that too may need to be explicitly rewarded. Such a change would help incorporate other forms of scholarship (e.g., software and data) and publicly oriented outputs (e.g., blog posts, policy briefs, podcasts), while, at the same time, promoting open access to all faculty work. Lastly, as mentioned above, a shift towards a more nuanced and judicious use of research metrics may allow for a greater number of activities, including those that are not readily quantifiable, to be considered and valued. Such a change is encouraged by the first of the principles in the Leiden Manifesto that states that "quantitative evaluation should support qualitative, expert assessment," not supplant it (*Hicks et al., 2015*).

While these suggestions may not fully align an institution's simultaneous goals of public good and academic productivity and output, we do believe that changing the guidelines and procedures governing the RPT process can have a significant impact on how faculty choose to allocate their time and energy. At the risk of putting

too much emphasis on one initiative (for a review of different initiatives, see *Moher et al., 2018*), we once again can point to the efforts of DORA in identifying good practices found in the documents of several research organizations (*DORA, 2018a*). However, for these and any other efforts to have the intended effect, more work continues to be needed to understand the relationship between RPT guidelines and faculty behavior (calls for such work go back as early as *O'Meara, 2002*).

The question therefore remains: do RPT guidelines truly influence faculty priorities and publishing strategies? Our analysis offers a glimpse of the extent to which various aspects of faculty work are present in formal guidelines, but it cannot tell us whether the presence of these terms, or the way they are used, is actually affecting how faculty spend their time, nor the successes and challenges they are finding through each activity. We believe further qualitative analysis of the sample of documents we collected, combined with surveys and interviews with faculty and RPT committees, could serve to explore the relationship that these documents have with the lived experience of RPT and to further understand how publicness intersects with faculty work. In the meantime, our work leads us to confirm that faculty are more often rewarded for publishing traditional research outputs and demonstrating that those outputs are cited by other scholars than for truly promoting public scholarship. As such, there is great potential to better align public scholarship goals with the metrics and RPT process that guides faculty work.

## Acknowledgements

The authors would like to acknowledge Kendal Crawford and Lisa Matthias for their relentless efforts to collect the documents, which turned out to be much more difficult than any of us envisioned. We would also like to thank Drs. Hannah McGregor and Jonathan Tennant for their helpful comments on an earlier draft of this manuscript, and the three thoughtful reviewers who similarly helped us improve the work. Finally, we would also like to thank and acknowledge the OpenCon community who brought us together in the first place, and whose work inspires and invigorates us year after year. May your openness be rewarded.

**Juan P Alperin** is in the School of Publishing and the Scholarly Communications Lab, Simon Fraser University, Vancouver, Canada

juan@alperin.ca

http://orcid.org/0000-0002-9344-7439

**Carol Muñoz Nieves** is in the Scholarly Communications Lab, Simon Fraser University, Vancouver, Canada

http://orcid.org/0000-0002-8857-3000

**Lesley A Schimanski** is in the Scholarly Communications Lab, Simon Fraser University, Vancouver, Canada

http://orcid.org/0000-0002-4664-179X

**Gustavo E Fischman** is at the Mary Lou Fulton Teachers College, Arizona State University, Tempe, United States

http://orcid.org/0000-0003-3853-9856

**Meredith T Niles** is in the Department of Nutrition and Food Sciences, University of Vermont, Burlington, United States

https://orcid.org/0000-0002-8323-1351

**Erin C McKiernan** is in the Departamento de Física, Universidad Nacional Autónoma de México, Mexico City, Mexico

https://orcid.org/0000-0002-9430-5221

*Author contributions:* Juan P Alperin, Conceptualization, Resources, Formal analysis, Supervision, Funding acquisition, Visualization, Methodology, Writing—original draft, Project administration, Writing—review and editing; Carol Muñoz Nieves, Data curation, Formal analysis, Writing—review and editing; Lesley A Schimanski, Conceptualization, Formal analysis, Supervision, Methodology, Writing—original draft, Project administration, Writing—review and editing; Gustavo E Fischman, Writing—original draft, Writing—review and editing; Meredith T Niles, Conceptualization, Funding acquisition, Methodology, Writing—review and editing; Erin C McKiernan, Conceptualization, Supervision, Funding acquisition, Methodology, Writing—review and editing

*Competing interests:* Erin C McKiernan: is a member of the DORA Steering Committee and an advisor for the Metrics Toolkit, both volunteer positions. The other authors declare that no competing interests exist.

## Funding

| Funder | Grant reference number | Author |
|---|---|---|
| Open Society Foundations | OR2016-29841 | Juan P Alperin Meredith T Niles Erin C McKiernan |

The funders had no role in study design, data collection and interpretation, or the decision to submit the work for publication.

**Decision letter and Author response**

Decision letter https://doi.org/10.7554/eLife.42254.017

Author response https://doi.org/10.7554/eLife.42254.018

## Additional files

### Supplementary files

• Transparent reporting form

DOI: https://doi.org/10.7554/eLife.42254.012

### Data availability

The data that support the findings of this study are available in the Harvard Dataverse with the identifier https://doi.org/10.7910/DVN/VY4TJE (Alperin et al., 2019). These data include the list of institutions and academic units for which we have acquired documents along with an indicator of whether each term and concept studied was found in the documents for the institution or academic unit. The data also include the aggregated values and chi-square calculations reported. The code used for computing these aggregations can be found on Github https://github.com/ScholCommLab/rpt-project (Alperin, 2019; copy archived at https://github.com/elifesciences-publications/rpt-project). The documents collected are available on request from the corresponding author (JPA). These documents are not publicly available due to copyright restrictions.

The following dataset was generated:

| Author(s) | Year | Dataset URL | Database and Identifier |
|---|---|---|---|
| Alperin JP, Muñoz Nieves C | 2018 | https://doi.org/10.7910/DVN/VY4TJE | Harvard Dataverse, 10.7910/DVN/VY4TJE |

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
