## [Decision Letter]

Thank you for submitting your article "How significant are the public dimensions of faculty work in review, promotion, and tenure documents?" for consideration by *eLife*. Your article has been reviewed by three peer reviewers, and the evaluation has been overseen by Emma Pewsey as the Reviewing Editor and Peter Rodgers as the Senior Editor. The following individuals involved in review of your submission have agreed to reveal their identity: Emily Janke (Reviewer #1); Heather Coates (Reviewer #2).

The reviewers have discussed the reviews with one another and I have drafted this decision to help you prepare a revised submission.

Summary:

This is an important and timely study that provides a unique perspective on how higher education interacts with, serves, and is viewed by the public. The study contributes an important data set and novel analysis techniques that contextualize previous studies, describe important characteristics and set a foundation for longitudinal analysis that could be incredibly valuable in capturing how review, promotion and tenure processes change over time. However, there are a number of concerns to be addressed regarding its analytical framework, analysis and discussion that will need to be addressed before publication.

Essential revisions:

Definition of terms

1) Please revise the text to be clear and consistent in your use of terms such as community, community engagement, public scholarship, and public engagements, and make sure these are all introduced in the Introduction section. This is important, particularly around the term “community engagement.” For example, in the subsection “Public and/or community engagement in research and scholarship,” the term community engagement is introduced, whereas previously, the term was community. This may confuse the reader who understands community and community engagement to be two different concepts. There is also a need to make a clear distinction between public good (as in public commodity) and public good (as in common good, altruistic mission etc.).

2) A good number of the institutions that are included in the study are institutions that have received the Carnegie Foundation's Community Engagement Elective Classification. In several of the quotes, the institutions were using definitions that were almost exactly the same as Carnegie's. Please make this clear in the text: this is relevant context because part of the application process for the elective classification asks about RPT and definitions of community engagement. Some institutions have used the Carnegie classification process as a way of informing and influencing revisions to RPT guidelines.

Methods and analysis

3) Please explain the rationale behind the choice to use a 15-word limit for the word frequency analysis.

4) It would be helpful for readers to have more detail about how NVivo was used to count mentions; at the very least, a description of the NVivo settings is necessary. If this information is too detailed for inclusion in the manuscript, it could be added to the documentation provided with the data.

5) With regard to the statement that the coding was refined using “all the associated variants identified by Barreno et al., 2013,” please describe these variants further (either here or at the earlier mention in the Introduction).

6) Public-status of the university should be used as a background variable, in addition to as a dependent variable. This could be done in two ways. First, universities could be classified in to two (or more) categories based on the public ownership. Second, universities could be classified as public and private (or several different categories) by the source of funding (public, private).

7) When analyzing the differences by disciplines the US and Canadian universities are considered to be part of the same population – please explain the rationale for this decision.

Connections with existing literature

8) Advocates of community-engaged scholarship have worked hard to clearly differentiate community-engaged scholarship from community service and public outreach – especially in RPT guidelines. Please include some reference to the body of scholarship on this topic. For example, over the past 20 years, we have seen various statements, “toolkits,” and standards for documenting and evaluating community-engaged scholarship in faculty RPT guidelines, including the Carnegie Foundation's Elective Community Engagement Classification application framework, the Association of Public Land-Grant Universities' Task Force on the New Engagement, Research University Engaged Scholarship Toolkit developed by The Research University Civic Engagement Network (TRUCEN), the Community-Engaged Scholarship Toolkit developed by Community-Campus Partnerships for Health, the Scholarship in Public: Knowledge Creation and Tenure Policy in the Engaged University developed by Imagining America, and a partnership of eight Canadian universities that developed Rewarding Community-Engaged Scholarship: Transforming University Policies and Practices.

Presentation of results

9) The unusual presentation of the chi-square results makes it difficult for the reader to locate important statistical information such as the degrees of freedom and the significance level for each result. I would recommend that the authors report the analytical results in a way that is consistent with standard practice (e.g., [https://depts.washington.edu/psych/files/writing_center/stats.pdf]). Please explicitly state that a chi-square was used to test independence for the results shown in Figures 1-3 and 6-8, along with the null and alternate hypotheses for each test. In each case, when stating that the conditions were not met for the chi-square test, please indicate what conditions were not met.

10) Please confirm the levels of significance shown in Figures 1 and 3: these are given 0.05, 0.01 and 0.001 in the subsection “Identification of terms,” but in the figures themselves are labelled 0.1, 0.05, 0.001. If 0.1 is correct, please explain why this has been included, as it is generally considered below the threshold for significance.

Discussion

11) The Discussion should be revised to alter the normative tone of the discussion and recommendation. Publicness is not a synonym for good and it is not especially a synonym for OA. Also traditional metrics can be considered to be a proxy of publicness of the university if we define the role of public universities as “seeking the objective truth” or as a following of “Mertonian norms of science” or as “Humboldtian universities.” The only way of providing normative policy advice for universities would be a systematic analysis of the strategies of the universities and the covariance of the RPT-documents with the strategy. Currently, the lack of this connection is more hypothetical than based on empirical findings.

---

## [Author Response]

Essential revisions:Definition of terms1) Please revise the text to be clear and consistent in your use of terms such as community, community engagement, public scholarship, and public engagements, and make sure these are all introduced in the Introduction section. This is important, particularly around the term “community engagement.” For example, in the subsection “Public and/or community engagement in research and scholarship,” the term community engagement is introduced, whereas previously, the term was community. This may confuse the reader who understands community and community engagement to be two different concepts. There is also a need to make a clear distinction between public good (as in public commodity) and public good (as in common good, altruistic mission etc.).

We addressed this point through a few changes:

First, we addressed it by being more explicit in the Introduction section when discussing community engaged scholarship (CES) and related practices. As we present in the Introduction, we see CES as one of the ways in which universities may try to address the public dimensions of faculty work (in particular, the “public good” dimension) and so bolstered the paragraph stating as much by acknowledging the growing body of work about CES and related practices, as well as referring to the statements and toolkits that the reviewers recommended.

More significantly, we have now been much more explicit about our choice of the term “public and/or community engagement” and how it is distinct from the terms “public” and “community.” In particular, we added all of the terms and further explained our strategy for coding the concept of “public and/or community engagement.” These additions can be seen at the beginning of the subsection “Public and/or community engagement in research and scholarship”. Reviewers will also notice a correction of the term in other places where we had relied on “community engagement” as a shorthand.

2) A good number of the institutions that are included in the study are institutions that have received the Carnegie Foundation's Community Engagement Elective Classification. In several of the quotes, the institutions were using definitions that were almost exactly the same as Carnegie's. Please make this clear in the text: this is relevant context because part of the application process for the elective classification asks about RPT and definitions of community engagement. Some institutions have used the Carnegie classification process as a way of informing and influencing revisions to RPT guidelines.

We agree that it is important to acknowledge the Carnegie Community Engagement classification, even if it is elective and therefore not available for all our units. We have now included it in two ways: First, by acknowledging its existence and influence in the introduction in the paragraph regarding community engagement. Second, by explicitly pointing out that some of the quotes draw very directly from this definition of community engagement. Lastly, we calculated the number of institutions that had opted into the classification and the number of them that received the designation, and included these figures in the text.

The relevant change can be seen in the subsection “Public and/or community engagement in research and scholarship” where we added: “This particular instance, and others like it, draw their definition of community engaged scholarship from the Carnegie classification described earlier, which explicitly asks if community engaged scholarship is rewarded in faculty promotion. In our sample, 85 institutions (all from the United States) had opted to have their community engagement assessed, and 34 had attained the classification.”

Methods and analysis3) Please explain the rationale behind the choice to use a 15-word limit for the word frequency analysis.

We are not aware of a specific standard distance used in linguistics research for such analysis, so we had opted for a word limit that approximated one sentence on either side of the term of interest. The average sentence length in English is estimated at 14.4 words (The Acropolitan, 2017). We have made this rationale explicit and included a citation in the first mention of searching “near” a term of interest (subsection “Context surrounding public and community”).

4) It would be helpful for readers to have more detail about how NVivo was used to count mentions; at the very least, a description of the NVivo settings is necessary. If this information is too detailed for inclusion in the manuscript, it could be added to the documentation provided with the data.

We included a line indicating that we used the “matrix query” option in NVivo to generate the incidence matrix (subsection “Identification of terms”, second paragraph) and specified that we used code written in Python to calculate numbers. We have now also published this code and cited it at the end of that same paragraph.

5) With regard to the statement that the coding was refined using “all the associated variants identified by Barreno et al., 2013,” please describe these variants further (either here or at the earlier mention in the Introduction).

This was done, as described above under point 1 in the subsection “Definitions of terms”.

6) Public-status of the university should be used as a background variable, in addition to as a dependent variable. This could be done in two ways. First, universities could be classified in to two (or more) categories based on the public ownership. Second, universities could be classified as public and private (or several different categories) by the source of funding (public, private).

We appreciate the reviewers' comments, but are not sure this distinction is significant in this case. In the Introduction, we mention how both public and private universities have public dimensions (e.g., private institutions receive public funds for research, and public institutions receive corporate donations), and agree with the analysis of Simon Marginson, one of the most influential Higher Education scholars, who argues that defining public/private universities in terms of legal ownership does not advance our comprehension in terms of the special characteristics of higher education as producers of knowledge. Instead Marginson suggests that we should focus on the social character of the goods produced by an organization when he argues “that public/private goods are not always zero sum and under certain conditions provide conditions of possibility for each other”. Perhaps a more important distinction here would be whether the institutions are profit or nonprofit, but in our sample, there are no for-profit colleges or universities to analyze.

Marginson, S. (2011). Higher education and public good. Higher Education Quarterly, 65(4), 411-433.

7) When analyzing the differences by disciplines the US and Canadian universities are considered to be part of the same population – please explain the rationale for this decision.

Given the similarity between the RPT processes in both countries, our preference throughout the analysis was to consider all the institutions as part of a single population. Such a treatment is commonly found in the literature (for a few examples see Goldstein and Bearman, 2011; Genshaft et al., 2016; Gruzd, Staves and Wilk, 2011). Where we have deviated from this practice is in the analysis of institution subtypes (i.e. R1, R2, R3 and RCan). The decision to keep the Canadian institutions separate was borne from the lack of commonly accepted subtype classification of the Canadian institutions. That is, we had no basis with which to classify Canadian R-types as being either R1, R2, or R3.

Genshaft, J., Wickert, J., Gray-Little, B., Hanson, K., Marchase, R., Schiffer, P., & Tanner, R. M. (2016). Consideration of Technology Transfer in Tenure and Promotion. Technology and Innovation, 197–20. https://doi.org/10.3727/194982416X14520374943103

Connections with existing literature8) Advocates of community-engaged scholarship have worked hard to clearly differentiate community-engaged scholarship from community service and public outreach – especially in RPT guidelines. Please include some reference to the body of scholarship on this topic. For example, over the past 20 years, we have seen various statements, “toolkits,” and standards for documenting and evaluating community-engaged scholarship in faculty RPT guidelines, including the Carnegie Foundation's Elective Community Engagement Classification application framework, the Association of Public Land-Grant Universities' Task Force on the New Engagement, Research University Engaged Scholarship Toolkit developed by The Research University Civic Engagement Network (TRUCEN), the Community-Engaged Scholarship Toolkit developed by Community-Campus Partnerships for Health, the Scholarship in Public: Knowledge Creation and Tenure Policy in the Engaged University developed by Imagining America, and a partnership of eight Canadian universities that developed Rewarding Community-Engaged Scholarship: Transforming University Policies and Practices.

We have brought in mentions of these statements and toolkits, especially the Carnegie Foundation’s Elective Community Classification, along with reference to a full bibliography on community engagement in addressing the earlier point regarding the definition and use of the terms “public,” “community,” and “public and/or community engagement.” While our work does not delve into or contribute to the fine distinctions made in the literature, we hope that our more careful treatment and use of the terms, along with the clearer links to the established definitions, address the underlying concern of the reviewers.

Presentation of results9) The unusual presentation of the chi-square results makes it difficult for the reader to locate important statistical information such as the degrees of freedom and the significance level for each result. I would recommend that the authors report the analytical results in a way that is consistent with standard practice (e.g., [https://depts.washington.edu/psych/files/writing_center/stats.pdf]). Please explicitly state that a chi-square was used to test independence for the results shown in Figures 1-3 and 6-8, along with the null and alternate hypotheses for each test. In each case, when stating that the conditions were not met for the chi-square test, please indicate what conditions were not met.

We have now added more careful reporting of the Chi-square analysis following the reviewers’ suggested guidelines. The additions can be seen at the end of the paragraphs following Figures 1, 3, 6, and 8.

10) Please confirm the levels of significance shown in Figures 1 and 3: these are given 0.05, 0.01 and 0.001 in the subsection “Identification of terms,” but in the figures themselves are labelled 0.1, 0.05, 0.001. If 0.1 is correct, please explain why this has been included, as it is generally considered below the threshold for significance.

Thank you for identifying this typo. The value used was 0.01, but was incorrectly written as 0.1 in the figure legend. These have been corrected.

Discussion11) The Discussion should be revised to alter the normative tone of the discussion and recommendation. Publicness is not a synonym for good and it is not especially a synonym for OA. Also traditional metrics can be considered to be a proxy of publicness of the university if we define the role of public universities as “seeking the objective truth” or as a following of “Mertonian norms of science” or as “Humboldtian universities.” The only way of providing normative policy advice for universities would be a systematic analysis of the strategies of the universities and the covariance of the RPT-documents with the strategy. Currently, the lack of this connection is more hypothetical than based on empirical findings.

Thank you for this suggestion. While we thought we had been careful to avoid such language, on rereading the Discussion we agreed with the reviewers about the normative tone. We have revised our Discussion section to remove instances of such language and to be more careful about how we frame our interpretations and our suggestions for what might be done.